# OpenReview forum: "Don’t Make It Up: Preserving Ignorance Awareness in LLM Fine-Tuning"
_NeurIPS.cc/2025/Workshop/Reliable_ML — NeurIPS 2025 - Reliable ML Workshop_

### Official Review · Reviewer_pVdg · 2025-09-10
**Preserving Ignorance Awareness in LLM Fine-Tuning with SEAT**

**Rating:** 7
**Confidence:** 4

**Review:**

This paper addresses an important and underexplored challenge in large language model (LLM) fine-tuning: the degradation of ignorance awareness—the model’s ability to appropriately admit lack of knowledge rather than hallucinate. The authors show that conventional fine-tuning methods (e.g., full fine-tuning, LoRA) blur the epistemic boundary between known and unknown data, causing models to lose this alignment property. To mitigate this, they propose Sparse Entity-aware Tuning (SEAT), which combines sparse parameter updates with an entity perturbation strategy to disentangle target knowledge from semantically similar but unseen data. Both theoretical analysis and extensive experiments on synthetic (PISTOL, TOFU) and real-world datasets (RWD) demonstrate that SEAT retains strong fine-tuning performance while substantially outperforming baselines in preserving ignorance awareness.

Strengths
* Novelty & Relevance: The paper formalizes the notion of ignorance awareness and highlights its erosion under standard fine-tuning. This framing is original and highly relevant for reliable ML with imperfect or private data.
* Methodological Soundness: SEAT is conceptually simple yet well-motivated, combining sparsity to constrain activation drift with entity perturbation to reduce knowledge entanglement. Both components are shown to be indispensable through ablation studies .
* Theoretical Backing: The analysis connecting sparse updates to bounded activation displacement provides a principled justification for why sparsity preserves epistemic behavior .
* Empirical Validation: Results across multiple models (Llama3-8B, Qwen2.5-7B) and datasets are convincing. Human evaluation (IDKHA) complements automatic metrics, showing >95% correct recognition of ignorance in unverifiable queries.
* Clarity: The paper is clearly written, with case studies and visualizations (e.g., PCA of activations) that make the arguments intuitive.

Limitations
* Baselines: While full fine-tuning and sparse fine-tuning are evaluated, other recent alignment-preserving methods (e.g., R-tuning, refusal-aware fine-tuning) are not compared. This limits the strength of the empirical claims.
* Scalability and Cost: The approach is tested on mid-sized models (up to 8B). It remains unclear how SEAT scales to larger models or multi-task continual learning settings.
* Entity Perturbation Design: The perturbation mechanism is relatively ad hoc (random substitutions). It would be useful to explore more principled strategies or test sensitivity to perturbation quality.
* Ethics Discussion: The ethical implications of preserving ignorance awareness are only briefly touched on. Given the paper’s motivation (safety in high-stakes applications), a more developed treatment of risks and tradeoffs would strengthen the impact.

---

### Official Review · Reviewer_ryVM · 2025-09-15
**Peer review of Don’t Make It Up: Preserving Ignorance Awareness in LLM Fine-Tuning**

**Rating:** 7
**Confidence:** 5

**Review:**

Summary:
The authors propose a new algorithm, SEAT, that fine-tunes a large language model to address the issue they describe as "Ignorance Awareness." This describes the self-awareness by a model that the answer to a given prompt is uncertain and there is not a definitive of which the model is aware. The algorithm seeks to solve this issue using SEAT which involves two primary steps: sparse fine-tuning and "entity perturbation" which addresses the knowledge entanglement problem. Through empirical analyses, SEAT is shown to be more effective than other methods and has consistency across different datasets.

Strengths:
The paper provides a novel solution to a major problem in LLMs. The problem of ignorance awareness is well-defined and a large empirical analysis was completed to test the method. There are strong results showing improvement across all metrics in most cases and gives a compelling case for the effectiveness of the proposed method.

Weaknesses:
The metrics used for evaluation could be better defined. While the paper's focus is on introducing the SEAT algorithm, the validity of the results are the justification for its use. This is especially a concern for IDK_SM and IDK_CS methods. These two metrics are subject to the underlying list of ignorance expressions which is not provided. If the list is not comprehensive enough, the results may be misleading.

Suggestions:
Statistical testing would ensure results are consistent on the general scale. Also, a better description of the metrics would better justify why they are valid measurements of performance.

Ethics:
While the method is well stated, the results can be seen as quite subjective. Either statistical testing, deeper justification for their validity, or transparency about what is considered a valid ignorance expression would strengthen the results. The footnote adds some clarity but is not fully transparent.